# Turning large language models into cognitive models

**Marcel Binz**
Max Planck Institute for Biological Cybernetics
Tübingen, Germany
`marcel.binz@tue.mpg.de`

**Eric Schulz**
Max Planck Institute for Biological Cybernetics
Tübingen, Germany
`eric.schulz@tue.mpg.de`

## Abstract

Large language models are powerful systems that excel at many tasks, ranging from translation to mathematical reasoning. Yet, at the same time, these models often show unhuman-like characteristics. In the present paper, we address this gap and ask whether large language models can be turned into cognitive models. We find that – after finetuning them on data from psychological experiments – these models offer accurate representations of human behavior, even outperforming traditional cognitive models in two decision-making domains. In addition, we show that their representations contain the information necessary to model behavior on the level of individual subjects. Finally, we demonstrate that finetuning on multiple tasks enables large language models to predict human behavior in a previously unseen task. Taken together, these results suggest that large, pre-trained models can be adapted to become models of human cognition, which opens up future research directions toward building more general cognitive models.

## 1 Introduction

Large language models are neural networks trained on vast amounts of data to predict the next token for a given text sequence (Brown et al., 2020). These models display many emergent abilities that were not anticipated by extrapolating the performance of smaller models (Wei et al., 2022). Their abilities are so impressive and far-reaching that some have argued that they even show sparks of general intelligence (Bubeck et al., 2023). We may currently witness one of the biggest revolutions in artificial intelligence, but the impact of modern language models is felt far beyond, permeating into education (Kasneci et al., 2023), medicine (Li et al., 2023), and the labor market (Eloundou et al., 2023).

In-context learning – the ability to extract information from a context and to use that information to improve the production of subsequent outputs – is one of the defining features of such models. It is through this mechanism that large language models are able to solve a variety of tasks, ranging from translation (Brown et al., 2020) to analogical reasoning (Webb et al., 2022). Previous work has shown that these models can even successfully navigate when they are placed into classic psychological experiments (Binz & Schulz, 2023; Shiffrin & Mitchell, 2023; Coda-Forno et al., 2023; Dasgupta et al., 2022; Hagendorff et al., 2022). To provide just one example, GPT-3 – an autoregressive language model designed by OpenAI (Brown et al., 2020) – outperformed human subjects in a sequential decision-making experiment that required to balance between exploitative and exploratory actions (Binz & Schulz, 2023).

Even though these models show human-like behavioral characteristics in some situations, this is not always the case (Ullman, 2023; Mitchell & Krakauer, 2023). In the sequential decision-making experiment mentioned above, for instance, GPT-3 relied heavily on exploitative strategies, while people applied a combination of elaborate exploration strategies (Wilson et al., 2014). Moreover, GPT-3 stopped improving after only a few trials, while people continued learning as the task progressed.

In the present paper, we investigate whether it is possible to fix the behavioral discrepancy between large language models and humans. To do so, we rely on the idea of finetuning on domain-specific data. This approach has been fruitful across a number of areas (Sanh et al., 2019; Ouyang et al.,

2022) and eventually led to the creation of the term *foundation models* (Bommasani et al., 2021) – models trained on broad data at scale and adapted to a wide range of downstream tasks. In the context of human cognition, such domain-specific data can be readily accessed by tapping the vast amount of behavioral studies that psychologists have conducted over the last century. We exploited this fact and extracted data sets for several behavioral paradigms which we then used to finetune a large language model.

We show that this approach can be used to create models that describe human behavior better than traditional cognitive models. We verify our result through extensive model simulations, which confirm that finetuned language models indeed show human-like behavioral characteristics. Furthermore, we find that embeddings obtained from such models contain the information necessary to capture individual differences. Finally, we highlight that a model finetuned on two tasks is capable of predicting human behavior on a third, hold-out task. Taken together, our work demonstrates that it is possible to align large language models with human behavior by finetuning them on data from psychological experiments. We believe that this could open up new opportunities to harvest the power of large language models to inform our theories of human cognition in the future.

## 2 METHODS

We tested whether it is possible to capture how people make decisions through finetuning a large language model. For our analyses, we relied on the *Large Language Model Meta AI*, or in short: LLaMA (Touvron et al., 2023). LLaMA is a family of state-of-the-art foundational large language models (with either 7B, 13B, 33B, or 65B parameters) that were trained on trillions of tokens coming from exclusively publicly available data sets. We focused on the largest of these models – the 65B parameter version – in our upcoming analyses. LLaMA is publicly available, meaning that researchers are provided with complete access to the network architecture including its pre-trained weights. We utilized this feature to extract embeddings for several cognitive tasks and then finetuned a linear layer on top of these embeddings to predict human choices (see Figure 1a for a visualization). We call the resulting class of models CENTaUR, in analogy to the mythical creature that is half human and half ungulate.

For our analyses, we first passed the corresponding text prompt (see Figure 1 for stylized examples and Appendix A.1 for exact prompts) through LLaMA. We then extracted the activations of the final hidden layer and repeated this procedure for each participant and trial. We fitted separate regularized logistic regression models on the standardized data via a maximum likelihood estimation. Final model performance was measured through the predictive log-likelihood on hold-out data obtained using a 100-fold cross-validation procedure. In each fold, we split the data into a training set (90%), a validation set (9%), and a test set (1%). The validation set was used to identify the parameter $\alpha$ that controls the strength of the $\ell_2$ regularization term using a grid search procedure. We considered discrete $\alpha$-values of [0, 0.0001, 0.0003, 0.001, 0.003, 0.01, 0.03, 0.1, 0.3, 1.0]. The optimization procedure was implemented in PYTORCH (Paszke et al., 2019) and used the default LBFGS optimizer (Liu & Nocedal, 1989). Data and code for our study are available through the following GitHub repository: `https://github.com/marcelbinz/CENTaUR`.

## 3 RESULTS

We considered two paradigms that have been extensively studied in the human decision-making literature for our analyses: *decisions from descriptions* (Kahneman & Tversky, 1972) and *decisions from experience* (Hertwig et al., 2004). In the former, a decision-maker is asked to choose between one of two hypothetical gambles like the ones shown in Figure 1b. Thus, for both options, there is complete information about outcome probabilities and their respective values. In contrast, the decisions from experience paradigm does not provide such explicit information. Instead, the decision-maker has to learn about outcome probabilities and their respective values from repeated interactions with the task as shown in Figure 1d. Importantly, this modification calls for a change in how an ideal decision-maker should approach such problems: it is not enough to merely exploit currently available knowledge anymore but also crucial to explore options that are unfamiliar (Schulz & Gershman, 2019).

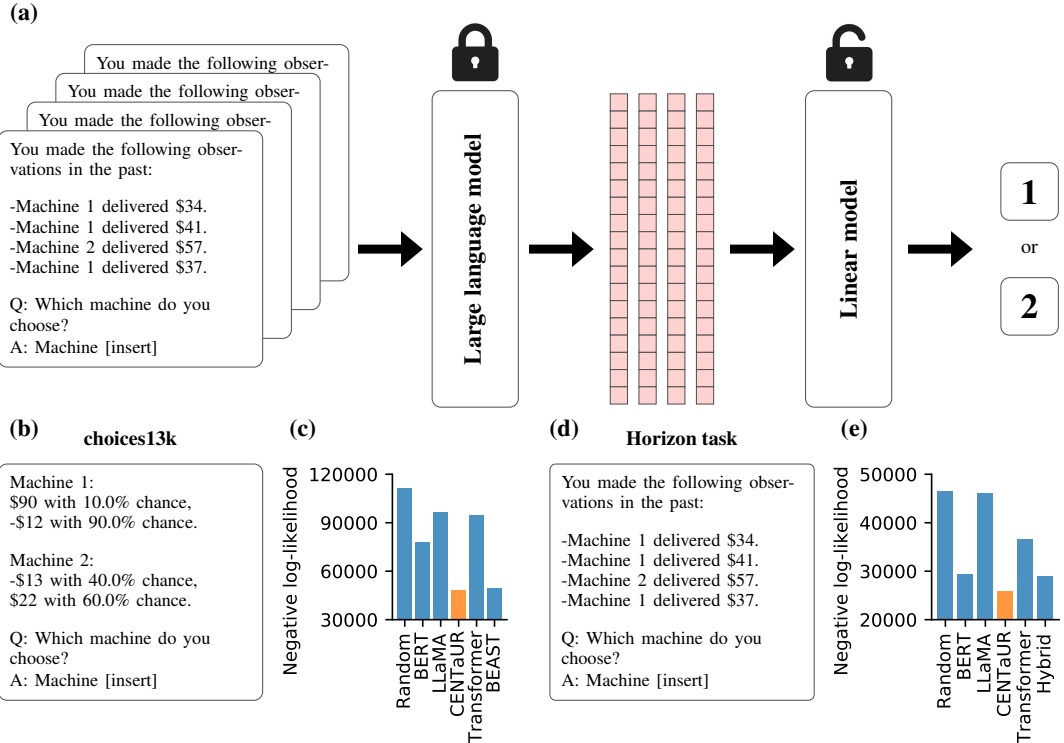

Figure 1: Illustration of our approach and main results. (a) We provided text-based descriptions of psychological experiments to a large language model and extracted the resulting embeddings. We then finetuned a linear layer on top of these embeddings to predict human choices. We refer to the resulting model as CENTaUR. (b) Example prompt for the choices13k data set. (c) Negative log-likelihoods for the choices13k data set. (d) Example prompt for the horizon task. (e) Negative log-likelihoods for the horizon task. Prompts shown in this figure are stylized for readability. Exact prompts can be found in Appendix A.1.

For both these paradigms, we created a data set consisting of embeddings and the corresponding human choices. We obtained embeddings by passing prompts that included all the information that people had access to on a given trial through LLaMA and then extracting the hidden activations of the final layer. We relied on publicly available data from earlier studies in this process. In the decisions from descriptions setting, we used the choices13k data set (Peterson et al., 2021), which is a large-scale data set consisting of over 13,000 choice problems (all in all, 14,711 participants made over one million choices on these problems). In the decisions from experience setting, we used data from the horizon task (Wilson et al., 2014) and a replication study (Feng et al., 2021), which combined include 60 participants making a total of 67,200 choices. The horizon task is a stationary two-armed bandit problem with normally-distributed rewards. Each task consists of either five or ten trials. The first four require the participant to select a predetermined option (forced-choice trials). They are followed by either one or six free-choice trials in which participants can freely decide between the two options. The number of free-choice trials is also known as the horizon. Participants are aware of the remaining choices and can use that information to guide their exploration behavior.

## 3.1 FINETUNED LANGUAGE MODELS BEAT DOMAIN-SPECIFIC MODELS

With these two data sets at hand, we fitted a regularized logistic regression model from the extracted embeddings to human choices as described above. In this section, we restricted ourselves to a joint model for all participants, thereby neglecting potential individual differences (but see one of the following sections for an analysis that allows for individual differences).

We compared the goodness-of-fit of the resulting models against five baselines (for further details see Appendix A.2):

1. a random guessing model.
2. a regularized logistic regression model with embeddings obtained from a simpler word embedding model (BERT, (Devlin et al., 2018)).
3. LLaMA without finetuning (obtained by reading out log-probabilities of the pre-trained model).
4. a regularized logistic regression model with embeddings obtained from a randomly initialized transformer (Transformer, (Schrimpf et al., 2021)).
5. a domain-specific model (*Best Estimate and Sampling Tools*, or BEAST, for the choices13k data set (Erev et al., 2017) and a *hybrid model* (Gershman, 2018) that involves a combination of different exploitation and exploration strategies for the horizon task).[1]

We found that LLaMA did not capture human behavior well, obtaining a negative log-likelihood (NLL) close to chance-level for the choices13k data set (NLL = 96248.5) and the horizon task (NLL = 46211.4). However, finetuning led to models that captured human behavior better than the domain-specific models under consideration. In the choices13k data set, CENTaUR achieved a negative log-likelihood of 48002.3 while BEAST only achieved a negative log-likelihood of 49448.1 (see Figure 1c). In the horizon task, CENTaUR achieved a negative log-likelihood of 25968.6 while the hybrid model only achieved a negative log-likelihood of 29042.5 (see Figure 1e). The two other baselines (BERT and Transformer) were both significantly better than chance but did not reach the level of CENTaUR. Figure 6 in Appendix A.3 further corroborates these results using accuracy as a measurement of goodness-of-fit.

To ensure that our fitting procedure is robust to random noise, we repeated this analysis with five different seeds. CENTaUR's results were almost identical across seeds with an average negative log-likelihood of 48002.3 (SE = 0.02) for the choices13k data set and an average negative log-likelihood of 25968.6 (SE = 0.02) for the horizon task. In both cases, these values were significantly lower than those of the domain-specific models ($p < 0.001$). Together, these results suggest that the representations extracted from large language models are rich enough to attain state-of-the-art results for modeling human decision-making.

## 3.2 MODEL SIMULATIONS REVEAL HUMAN-LIKE BEHAVIOR

We next verified that CENTaUR shows human-like behavioral characteristics. To do so, we simulated the model on the experimental tasks. Looking at performance, we found that finetuning led to models that closely resemble human performance as shown in Figure 2a and b. For the choices-13k data set, CENTaUR obtained a regret (defined as the difference between the highest possible reward and the reward for the action selected by the model) of 1.35 (SE = 0.01), which was much closer to the human regret (M = 1.24, SE = 0.01) than the regret of LLaMA (M = 1.85, SE = 0.01). The results for the horizon task showed an identical pattern, with CENTaUR (M = 2.38, SE = 0.01) matching human regret (M = 2.33, SE = 0.05) more closely than LLaMA (M = 7.21, SE = 0.02).

In addition to looking at performance, we also inspected choice curves. For this analysis, we took the data from the first free-choice trial in the horizon task and divided it into two conditions: (1) an equal information condition that includes trials where the decision-maker had access to an equal number of observations for both options and (2) an unequal information condition that includes trials where the decision-maker previously observed one option fewer times than the other. We then fitted a separate logistic regression model for each condition with reward difference, horizon, and their interaction as independent variables onto the simulated choices. Earlier studies with human subjects (Wilson et al., 2014) identified the following two main results regarding their exploratory behavior: (1) people's choices become more random with a longer horizon in the equal information condition (as shown in Figure 2c) and (2) people in the unequal information condition select the more informative option more frequently when the task horizon is longer (as shown in Figure 2d). While LLaMA did not show any of the two effects (see Figure 2e and f), CENTaUR exhibited both of them (see Figure 2g and h), thereby further corroborating that it accurately captures human behavior.

---

[1]There are a number of other domain-specific models that could be used as baselines instead (Bourgin et al., 2019; He et al., 2022; Zhang & Yu, 2013; Wilson et al., 2014). While the ones we selected are good representatives of the literature, future work should strive to conduct a more exhaustive comparison.

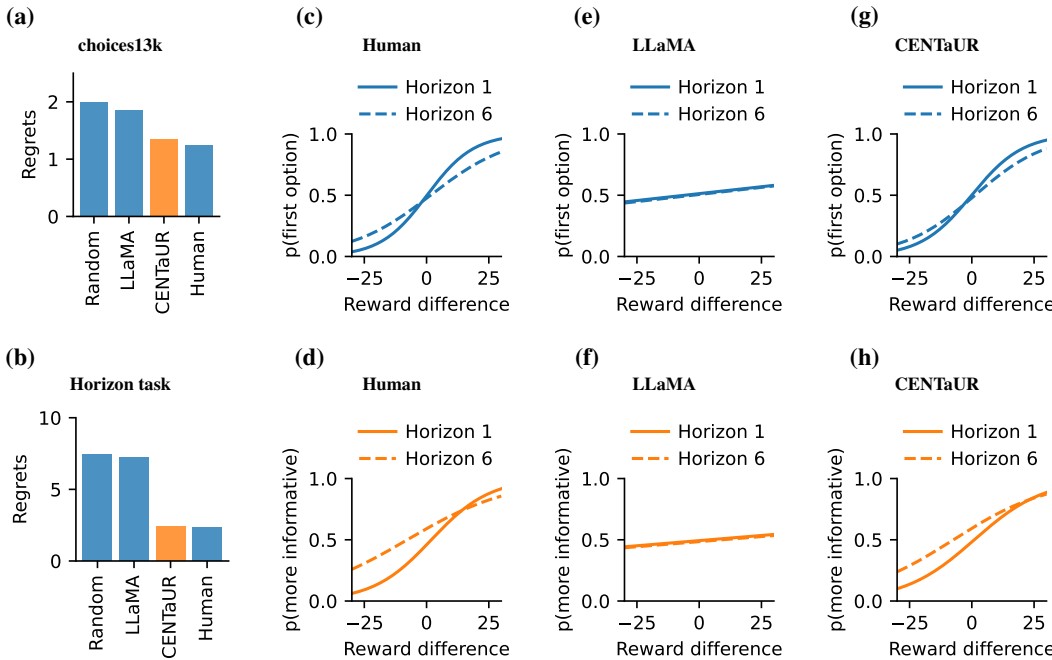

Figure 2: Model simulations. (a) Performance for different models and human participants on the choices13k data set. (b) Performance for different models and human participants on the horizon task. (c) Human choice curves in the equal information condition of the horizon task. (d) Human choice curves in the unequal information condition of the horizon task. (e) LLaMA choice curves in the equal information condition of the horizon task. (f) LLaMA choice curves in the unequal information condition of the horizon task. (g) CENTaUR choice curves in the equal information condition of the horizon task. (h) CENTaUR choice curves in the unequal information condition of the horizon task.

### 3.3 LANGUAGE MODEL EMBEDDINGS CAPTURE INDIVIDUAL DIFFERENCES

We also investigated how well CENTaUR describes the behavior of each individual participant. Note that this form of analysis is only possible for the horizon task as choice information on the participant level is not available for the choices13k data set. In total, the majority of participants (N = 52 out of 60) was best modeled by CENTaUR (see Figure 3a for a detailed visualization). We furthermore entered the negative log-likelihoods into a random-effects model selection procedure which estimates the probability that a particular model is the most frequent explanation within a set of candidate models (Rigoux et al., 2014). This procedure favored CENTaUR decisively, assigning a probability that it is the most frequent explanation of close to one.

Thus far, we have finetuned LLaMA jointly for all participants. However, people may exhibit individual differences that are not captured by this analysis. To close this gap and test whether LLaMA embeddings can account for individual differences, we incorporated random effects in the finetuned layer. We added a random effect for each participant and embedding dimension while keeping the remaining evaluation procedure the same. Figure 3b illustrates the resulting negative log-likelihoods. The mixed-effect structure improved goodness-of-fit considerably (NLL = 23929.5) compared to the fixed-effect-only model (NLL = 25968.6). Furthermore, CENTaUR remained superior to the hybrid model with an identical mixed-effect structure (NLL = 24166.0). Taken together, the findings reported in this section highlight that embeddings of large language models contain the information necessary to model behavior on the participant level.

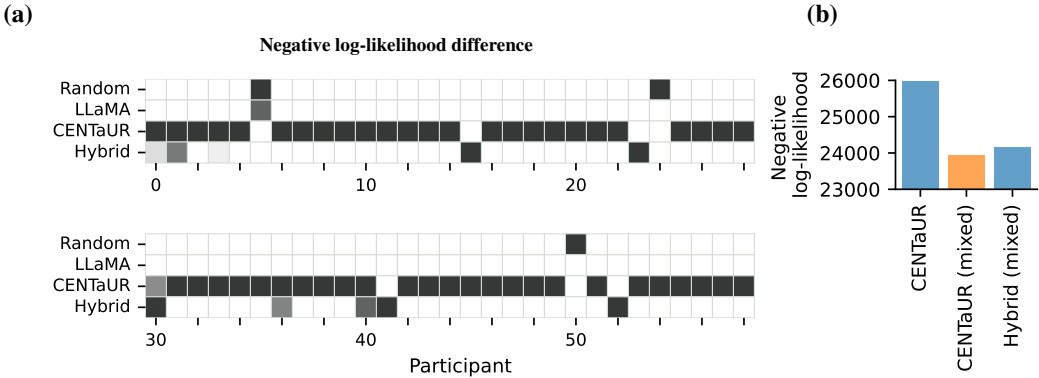

Figure 3: Individual differences. (a) Negative log-likelihood difference to the best-fitting model for each participant. Black coloring highlights the best-fitting model, while white coloring corresponds to a difference larger than ten. (b) Negative log-likelihoods for models that were finetuned using the mixed-effects structure described in the main text.

## 3.4 EVALUATING GOODNESS-OF-FIT ON HOLD-OUT TASKS

Next, we examined whether CENTaUR – after being finetuned on multiple tasks – is able to predict human behavior in an entirely different task. This evaluation protocol provides a much stronger test for the generalization abilities of our approach. Following our initial analyses, we finetuned a linear layer on top of LLaMA embeddings. However, this time, we fitted a joint model using both the data from the choices13k data set and the horizon task, and then evaluated how well the finetuned model captures human choices on a third task. Further details about the fitting procedure are provided in Appendix A.4. For the hold-out task, we considered data from a recent study that provided participants with a choice between one option whose information is provided via a description and another option for which information is provided via a list of experienced outcomes (Garcia et al., 2023). Figure 4a shows an example prompt for this experimental paradigm. In total, the experiment involved 98 participants making 25,872 choices. We only considered the mixed trials of the post-learning phase for our analysis, leading to a data set size of 8,624.

Finetuning was generally beneficial for modeling human behavior on the hold-out task: negative log-likelihoods for CENTaUR (NLL = 4521.1) decreased both in comparison to a random guessing model (NLL = 5977.7) and LLaMA (NLL = 6307.9). We were thus curious whether CENTaUR also captures human behavior on a qualitative level. To test this, we took a look at the key insight from the original study: people tend to overvalue options that are provided through a description (known as symbolic or S-options) over the options that come with a list of experienced outcomes (known as experiential or E-options) as illustrated in Figure 4b and c. LLaMA does not show this characteristic and instead weighs both option types equally (Figure 4d and e). In contrast to this, CENTaUR shows human-like behavior, taking the S-option more strongly into account (Figure 4f and g). This is especially remarkable because we never presented data from the experiment under consideration during finetuning.

## 3.5 USING CENTaUR TO INFORM COGNITIVE THEORIES

CENTaUR may also be used to gain insights into the cognitive processes that underlie observed behaviors. For instance, it can be used as a benchmark against which traditional cognitive models can be compared. This can help us to identify behavioral patterns that are predictable but that are not captured by existing cognitive models (Agrawal et al., 2020). In the following, we present a proof of concept analysis in which we have applied this idea to the two paradigms considered in this paper.

We first computed differences between the log-likelihoods of CENTaUR and the domain-specific model (BEAST for the choices13k data set and the hybrid model for the horizon task) for each data point. We then ranked the data points and inspected those with the largest log-likelihood difference.

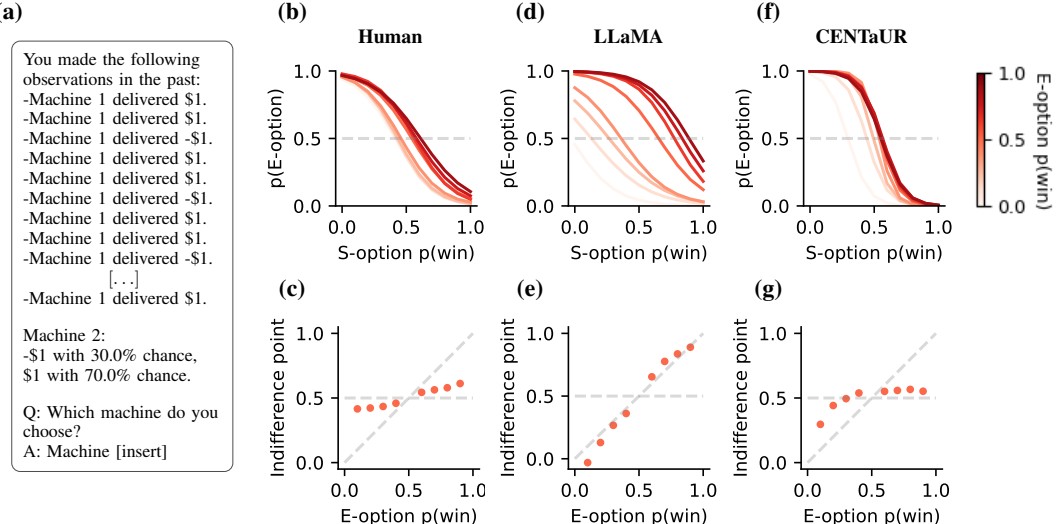

Figure 4: Hold-out task evaluations. (a) Example prompt for the experiential-symbolic task of Garcia et al. (2023). (b) Human choice curves as a function of win probabilities for both options. (c) Human indifference points as a function of win probability for the E-option. Indifferent points express the win probabilities at which a decision-maker is equally likely to select both options. (d) LLaMA choice curves as a function of win probabilities for both options. (e) LLaMA indifference points as a function of win probability for the E-option. (f) CENTaUR choice curves as a function of win probabilities for both options. (g) CENTaUR indifference points as a function of win probability for the E-option.

These points correspond to cases where CENTaUR accurately predicted human behavior but the domain-specific model did not. Figure 5 shows three examples for each paradigm.

For the choices13k data set, we observed that the data points with the largest log-likelihood difference correspond to problems where (1) one option consists of purely negative outcomes, (2) the other option contains at least one positive outcome, and (3) the expected value of the purely negative option is higher. Humans, like CENTaUR, prefer the option with the positive outcome, whereas BEAST does not. We interpret this as a form of loss aversion (Kahneman, 1979) that is context-dependent. This bias could be integrated into the BEAST model to further advance our theories of human decision-making.

For the horizon task, we observed that the data points with the largest log-likelihood difference correspond to problems with (1) a high mean reward difference between options, (2) a long horizon, and (3) where people have repeatedly selected the option with the lower mean value. CENTaUR is able to capture that people will stick to the less valuable option while the hybrid model is not, suggesting that the hybrid model could be improved by adding a stickiness component (Gershman, 2020; Jagadish et al., 2023).

## 4 DISCUSSION

We have demonstrated that large language models can be turned into cognitive models by finetuning their final layer. This process led to models that outperformed traditional cognitive models in their goodness-of-fit to human behavior. Furthermore, these models were able to capture behavioral differences at the individual participant level. Finally, we have shown that our approach generalizes to previously unseen tasks. In particular, a model that was finetuned on two tasks also exhibited human-like behavior on a third, hold-out task.

These results complement earlier work showing that large language model embeddings allow us to predict behavior and neural activations in linguistic settings (Schrimpf et al., 2021; Kumar et al., 2022; Tuckute et al., 2023; Antonello et al., 2023). For example, Schrimpf et al. (2021) showed that

**(a)**

| | | |
|---|---|---|
| Machine 1 delivers
  2 dollars with 75% chance,
-14 dollars with 25% chance.

Machine 2 delivers
-3.5 dollars with 12.5% chance,
-2.5 dollars with 37.5% chance,
-1.5 dollars with 37.5% chance,
-0.5 dollars with 12.5% chance.



Your goal is to maximize the amount of received dollars.

Q: Which machine do you choose?
A: Machine **1** | Machine 1 delivers
-6 dollars with 100% chance.

Machine 2 delivers
 -33 dollars with 50% chance,
-2.5 dollars with 1.5625% chance,
-1.5 dollars with 7.8125% chance,
-0.5 dollars with 15.625% chance,
 0.5 dollars with 15.625% chance,
 1.5 dollars with 7.8125% chance,
 2.5 dollars with 1.5625% chance.

Your goal is to maximize the amount of received dollars.

Q: Which machine do you choose?
A: Machine **2** | Machine 1 delivers
-9 dollars with 100% chance.

Machine 2 delivers
-41 dollars with 40% chance,
 -5 dollars with 30% chance,
 -3 dollars with 15% chance,
 1 dollars with 7.5% chance,
 9 dollars with 3.75% chance,
 25 dollars with 1.875% chance,
 57 dollars with 0.9375% chance,
121 dollars with 0.9375% chance.

Your goal is to maximize the amount of received dollars.

Q: Which machine do you choose?
A: Machine **2** |

**(b)**

| | | |
|---|---|---|
| You made the following observations in the past:
- Machine 2 delivered 34 dollars.
- Machine 1 delivered 62 dollars.
- Machine 1 delivered 64 dollars.
- Machine 2 delivered 30 dollars.
- Machine 1 delivered 59 dollars.
- Machine 2 delivered 39 dollars.
- Machine 2 delivered 26 dollars.
- Machine 2 delivered 16 dollars.
- Machine 2 delivered 12 dollars.

Your goal is to maximize the sum of received dollars within one additional choice.

Q: Which machine do you choose?
A: Machine **2** | You made the following observations in the past:
- Machine 2 delivered 74 dollars.
- Machine 2 delivered 75 dollars.
- Machine 1 delivered 52 dollars.
- Machine 1 delivered 40 dollars.
- Machine 2 delivered 88 dollars.
- Machine 2 delivered 74 dollars.
- Machine 2 delivered 70 dollars.
- Machine 1 delivered 34 dollars.
- Machine 1 delivered 45 dollars.

Your goal is to maximize the sum of received dollars within one additional choice.

Q: Which machine do you choose?
A: Machine **1** | You made the following observations in the past:
- Machine 1 delivered 50 dollars.
- Machine 2 delivered  2 dollars.
- Machine 2 delivered  2 dollars.
- Machine 1 delivered 39 dollars.
- Machine 2 delivered  8 dollars.
- Machine 2 delivered 15 dollars.
- Machine 2 delivered  1 dollars.
- Machine 2 delivered 20 dollars.
- Machine 2 delivered  4 dollars.

Your goal is to maximize the sum of received dollars within one additional choice.

Q: Which machine do you choose?
A: Machine **2** |

Figure 5: Prompts with the largest difference between the log-likelihoods of CENTaUR and the domain-specific model. Human choices are highlighted in green. CENTaUR matched human choices, while the domain-specific model did not. (a) Examples from the choices13k data sets. (b) Examples from the horizon task. Formatting adjusted for readability.

large language models can predict neural and behavioral responses in tasks that involved reading short passages with an accuracy that was close to noise ceiling. While it may be expected that large language models explain human behavior in linguistic domains (after all these models are trained to predict future word occurrences), the observation that these results also transfer to more cognitive domains like the ones studied here is highly non-trivial and has not been demonstrated before.

We are particularly excited about one feature of CENTaUR: embeddings extracted for different tasks all lie in a common space. This property allows finetuned large language models to solve multiple tasks in a unified architecture. We have presented preliminary results in this direction, showing that a model finetuned on two tasks can predict human behavior on a third. However, we believe that our current results only hint at the potential of this approach. Ideally, we would like to scale up our approach to finetuning on a larger number of tasks from the psychology literature. If one would include enough tasks in the training set, the resulting system should – in principle – generalize to *any* hold-out task. Therefore, our approach provides a path towards a domain-general model of human cognition, which has been the goal of psychologists for decades (Newell, 1992; Yang et al., 2019; Riveland & Pouget, 2022; Binz et al., 2023).

We believe that having access to such a domain-general model would transform psychology and the behavioral sciences more generally. Recently, researchers even started to wonder whether large language models can replace human participants in experimental studies (Dillion et al., 2023), i.e., whether they can be used as proxies for human behavior (Aher et al., 2023; Jiang et al., 2022). While we would not go as far, there are certainly compelling arguments for why to rely on such agents for prototyping: they are quicker, cheaper, and less noisy. Imagine, for instance, that you have just designed a new experimental paradigm. In a typical workflow, you would often run one or multiple pilot studies to gauge whether everything works as intended. However, with an accurate

domain-general model at hand, you may conduct the piloting step in silico, thereby saving time and money. In another situation, you may have an application that benefits from predicting how people act. For such use cases, a domain-general model would be again tremendously useful. Instead of running expensive data collection procedures, you could work directly with the simulated data of the model, which could enable many applications that were previously out of reach.

However, it is important to mention that the present paper still falls short of this goal. We would expect a generalist cognitive model to be robust to modifications in the prompt structure. While this is generally true for people, the same does not always hold for large language models (Frank, 2023a; Mitchell, 2023). For instance, a recent paper found that modern language models can solve 90% of false-belief tasks, thereby indicating theory of mind-like abilities (Kosinski, 2023). Follow-up papers, on the other hand, showed that performance breaks down as soon as the task structure is modified in trivial ways, suggesting that the evidence is not as clear-cut (Ullman, 2023; Shapira et al., 2023). We also found this to be the case for CENTaUR: performance breaks down once we evaluated finetuned models on tasks with slightly alternated prompt structures (see Appendix A.5 for further details). Identifying if and how these issues can be resolved is an important area for future research. We imagine that using data augmentation techniques and training the models on larger data sets can improve their robustness, but it is still unclear to what extent.

Finally, we have to ask ourselves what we can learn about human cognition when finetuning large language models. We have presented a proof of concept analysis demonstrating that CENTaUR can be used to inform cognitive theories. While this analysis is interesting in its own right, it is certainly not the end of the story. Looking beyond the current work, having access to an accurate neural network model of human behavior provides the opportunity to apply a wide range of explainability techniques from the machine learning literature. For instance, we could pick a particular neuron in the embedding and trace back what parts of an input sequence excite that neuron using methods such as layer-wise relevance propagation (Bach et al., 2015; Chefer et al., 2021). The obtained insights could be then fed back into traditional cognitive models, thereby improving their ability to characterize human behavior. It should be noted that analyses like this are not possible when working with human subjects alone and thus our work opens up a totally new spectrum of techniques that could be applied to study human cognition.

To summarize, large language models are an immensely powerful tool for studying human behavior (Demszky et al., 2023; Frank, 2023b). We believe that our work has only scratched the surface of this potential and there is certainly much more to come. Many questions only wait to be explored: are larger models generally more suitable for finetuning? How do techniques such as reinforcement learning from human feedback (Ziegler et al., 2019) influence the goodness-of-fit to human behavior? Is it possible to finetune entire networks instead of just their final layer using techniques such as low-rank adaptation (Hu et al., 2021)? However, the most crucial aspect that still ought to be studied in more detail is how far finetuned models are able to generalize beyond their training distribution.

### ACKNOWLEDGMENTS

This work was funded by the Max Planck Society, the Volkswagen Foundation, as well as the Deutsche Forschungsgemeinschaft (DFG, German Research Foundation) under Germany's Excellence Strategy–EXC2064/1–390727645.

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

## A  APPENDIX

### A.1  EXAMPLE PROMPTS

For the choices13k data set, we prompted each decision independently, thereby ignoring the potential effect of feedback. We used the following template:

> Machine 1 delivers 90 dollars with 10.0% chance and -12 dollars with 90.0% chance.
> Machine 2 delivers -13 dollars with 40.0% chance and 22 dollars with 60.0% chance.
>
> Your goal is to maximize the amount of received dollars.
>
> Q: Which machine do you choose?
> A: Machine [insert]

For the horizon task, we prompted each task independently, thereby ignoring potential learning effects across the experiment. We used the following template:

> You made the following observations in the past:
> - Machine 1 delivered 34 dollars.
> - Machine 1 delivered 41 dollars.
> - Machine 2 delivered 57 dollars.
> - Machine 1 delivered 37 dollars.
>
> Your goal is to maximize the sum of received dollars within six additional choices.
>
> Q: Which machine do you choose?
> A: Machine [insert]

For the experiential-symbolic task, we prompted each decision independently and only considered the post-learning phase. We furthermore simplified the observation history by only including the option that is relevant to the current decision. We used the following template:

> You made the following observations in the past:
> - Machine 1 delivered 1 dollars.
> - Machine 1 delivered 1 dollars.
> - Machine 1 delivered -1 dollars.
> - Machine 1 delivered 1 dollars.
> - Machine 1 delivered 1 dollars.
> - Machine 1 delivered -1 dollars.
>                    [. . .]
> - Machine 1 delivered 1 dollars.
>
> Machine 2 delivers -1 dollars with 30.0% chance and 1 dollars with 70.0% chance.
>
> Your goal is to maximize the amount of received dollars.
>
> Q: Which machine do you choose?
> A: Machine [insert]

Note that the "[insert]" token is shown only for visual purposes and not part of the actual prompt.

## A.2 BASELINE MODELS

For the LLaMA baseline, we passed the corresponding text prompt through LLaMA. We then extracted the unnormalized probabilities for the two options (i.e., tokens "1" and "2") and repeated this procedure for each participant and trial. We multiplied the unnormalized probabilities with a temperature parameter, which was fitted to human choices using the procedure described in the main text.

The fitting procedure for the BERT and Transformer baselines mirrored those of CENTaUR. We extracted BERT embeddings using the transformers library (Wolf et al., 2019). We extracted embeddings for the Transformer baseline by resetting LLaMA weights to their initial values before querying the model.

For the BEAST baseline, we relied on the version provided for the choice prediction competition 2018 (Plonsky et al., 2018). We additionally included an error model that selects a random choice with a particular probability. We treated this probability as a free parameter and fitted it using the procedure described in the main text. The hybrid model closely followed the implementation of Gershman (2018). We replaced the probit link function with a logit link function to ensure comparability to CENTaUR.

## A.3 ACCURACY RESULTS

Figure 6 contains a validation of our main analysis using accuracy as a measurement of goodness-of-fit. We again found that CENTaUR was the best performing model, achieving accuracies of 65.18% for the choices13k data set and 83.5% for the horizon task.[2] We also included GPT-4 (OpenAI, 2023) as a baseline in this analysis. GPT-4 performed worse when compared to CENTaUR for both the choices13k data set (59.72%) and the horizon task (80.3%) but better than LLaMA without any finetuning.

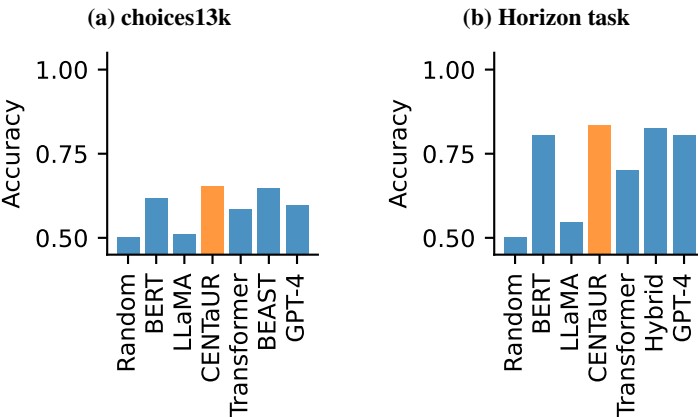

Figure 6: Replication of our main analysis (Figure 1) using accuracy as a measurement of goodness-of-fit.

## A.4 HOLD-OUT TASK ANALYSES

The training set consisted of the concatenated choices13k and horizon task data. To obtain a validation and test set, we split the data of the experiential-symbolic task into eight folds and repeated the previously described fitting procedure for each fold. The validation set was used to identify the

---

[2]Note that the accuracy differs from the decision preference accuracy reported by (Peterson et al., 2021). The decision preference accuracy is defined as the proportion of problems in which the model prediction for $P(A)$ is $> 0.5$ when the observed proportions are also $> 0.5$. CENTaUR achieves a decision preference accuracy of 81.9% on the choices13k data set.

parameter $\alpha$ that controls the strength of the $\ell_2$ regularization term and an inverse temperature parameter. We considered discrete inverse temperature values of [0.05, 0.1, 0.15, 0.2, 0.25, 0.3, 0.35, 0.4, 0.45, 0.5, 0.55, 0.6, 0.65, 0.7, 0.75, 0.8, 0.85, 0.9, 0.95, 1] and $\alpha$-values as described in the main text.

For our model simulations, we simulated data deterministically based on a median threshold (again using the predictions on the test set). The resulting choice curves were generated by fitting a separate logistic regression model for each possible win probability of the E-option. Each model used the win probability of the S-option and an intercept term as independent variables and the probability of choosing the E-option as the dependent variable.

### A.5 ROBUSTNESS CHECKS

We tested how robust CENTaUR is to minor modifications in the prompt structure. For this analysis, we finetuned CENTaUR on the choices13k data set as described in the main text but evaluated it on modified prompts. In particular, we considered the following two modifications:

1. Moving the instructions ("Your goal is to maximize the amount of received dollars.") and the question ("Q: Which machine do you choose?") to the beginning of the prompt.

2. Exchanging the order of probabilities and outcomes (i.e., "Machine 1 delivers 90 dollars with 10.0% chance and -12 dollars with 90.0% chance" becomes "Machine 1 has 10.0% chance to deliver 90 dollars and 90.0% chance to deliver -12 dollars.")

In general, we found that all of these modifications lead to a goodness-of-fit below chance level. The exact results of the analysis are shown in Figure 7.

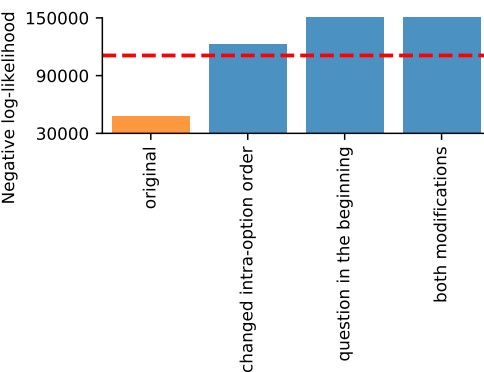

Figure 7: Robustness to minor prompt modifications on the choices13k data set. The dotted red line corresponds to the goodness-of-fit of a random guessing model.

## A.6 USING COGNITIVE THEORIES TO INFORM CENTAUR

We have also conducted an inverse of the analysis presented in Section 3.5. For this analysis, we inspected the data points with the lowest log-likelihood difference between CENTaUR and the domain-specific model. These points correspond to cases where the domain-specific model accurately predicted human behavior but CENTaUR did not. Figure 8 shows three examples for each paradigm.

For the choices13k data set, we were not able to identify a clear pattern in these data points. For the horizon task, we observed that the data points with the lowest log-likelihood difference correspond to cases where participants switched between choices in the last trial after repeatedly selecting the other option. We could not identify a clear pattern that would indicate this switch.

**(a)**

| | | |
|---|---|---|
| Machine 1 delivers
74 dollars with 1% chance,
10 dollars with 99% chance.

Machine 2 delivers
15 dollars with 95% chance,
27 dollars with 5% chance. | Machine 1 delivers
90 dollars with 50% chance,
-40 dollars with 50% chance.

Machine 2 delivers
0 dollars with 10% chance,
19 dollars with 90% chance. | Machine 1 delivers
47 dollars with 1% chance,
30 dollars with 99% chance.

Machine 2 delivers
-3 dollars with 1% chance,
26.5 dollars with 3.0938% chance,
27.5 dollars with 15.4688% chance,
28.5 dollars with 30.9375% chance,
29.5 dollars with 30.9375% chance,
30.5 dollars with 15.4688% chance,
31.5 dollars with 3.0938% chance. |
| Your goal is to maximize the amount of received dollars.

Q: Which machine do you choose?
A: Machine **1** | Your goal is to maximize the amount of received dollars.

Q: Which machine do you choose?
A: Machine **2** | Your goal is to maximize the amount of received dollars.

Q: Which machine do you choose?
A: Machine **1** |

**(b)**

| | | |
|---|---|---|
| You made the following observations in the past:
- Machine 1 delivered 67 dollars.
- Machine 1 delivered 61 dollars.
- Machine 2 delivered 63 dollars.
- Machine 2 delivered 65 dollars.
- Machine 2 delivered 68 dollars.
- Machine 2 delivered 67 dollars.
- Machine 2 delivered 65 dollars.
- Machine 2 delivered 79 dollars.
- Machine 2 delivered 72 dollars. | You made the following observations in the past:
- Machine 1 delivered 47 dollars.
- Machine 2 delivered 40 dollars.
- Machine 2 delivered 38 dollars.
- Machine 2 delivered 36 dollars.
- Machine 1 delivered 44 dollars.
- Machine 1 delivered 47 dollars.
- Machine 1 delivered 45 dollars.
- Machine 1 delivered 46 dollars.
- Machine 1 delivered 46 dollars. | You made the following observations in the past:
- Machine 2 delivered 77 dollars.
- Machine 2 delivered 77 dollars.
- Machine 1 delivered 59 dollars.
- Machine 1 delivered 58 dollars.
- Machine 1 delivered 67 dollars.
- Machine 1 delivered 63 dollars.
- Machine 1 delivered 48 dollars.
- Machine 1 delivered 71 dollars.
- Machine 1 delivered 46 dollars. |
| Your goal is to maximize the sum of received dollars within one additional choice.

Q: Which machine do you choose?
A: Machine **1** | Your goal is to maximize the sum of received dollars within one additional choice.

Q: Which machine do you choose?
A: Machine **2** | Your goal is to maximize the sum of received dollars within one additional choice.

Q: Which machine do you choose?
A: Machine **2** |

Figure 8: Prompts with the lowest difference between the log-likelihoods of CENTaUR and the domain-specific model. Human choices are highlighted in green. (a) Examples from the choices13k data sets. (b) Examples from the horizon task. Formatting adjusted for readability.

## A.7 EMBEDDING ANALYSIS

We performed a simple t-SNE (Van der Maaten & Hinton, 2008) analysis to investigate the information contained in the LLaMA embeddings. Figure 9 shows the results of this analysis. We found that the different tasks can be separated in the t-SNE space and that the embeddings for the experiential-symbolic task lie in between those of the two individual tasks. While the t-SNE space also displays some structure with regard to the difference in values between both options, the overall picture is less clear in this case.

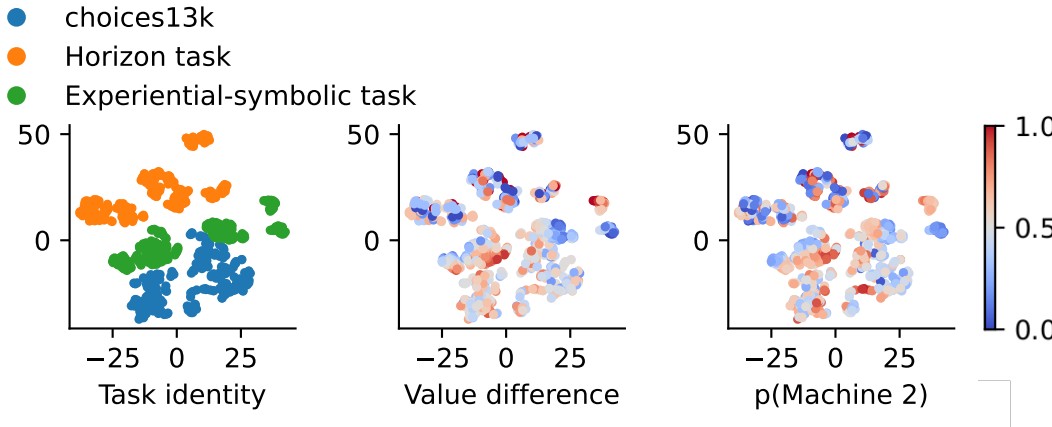

Figure 9: t-SNE analysis of LLaMA embeddings. The left plot shows 256 data points for each task with their task identity. The middle plot shows the same data points colored based on their normalized value difference (Machine 2 minus Machine 1). The right plot shows the same data points colored based on CENTaUR's probability of choosing Machine 2.

