# OpenReview forum: "Turning large language models into cognitive models"
_ICLR.cc/2024/Conference — ICLR 2024 poster_

### Official Review · Reviewer_donP · 2023-10-22

**Soundness:** 3 good
**Presentation:** 2 fair
**Contribution:** 3 good
**Rating:** 8
**Confidence:** 3

**Summary:**

Can language models capture aspects of human behaviour? In this work, these authors take a step towards this question by introducing fine-tuning an LLM (specifically LLaMA) on human behavioural data. The authors find that the resulting model – CENTaUR – is able to well-capture binary human decisions and achieves competitive performance against alternative cognitive models. In particular, the authors find that CENTaUR’s learned embeddings generalise well to new human behaviour data, raising the prospects of leveraging (finetuned) LLMs as stimuli-computable cognitive models.

**Strengths:**

The method proposed by the authors is sweet and simple (in a good way!). The idea of finetuning on human choice data is neat and impressively improves fits to human data. I believe the CENTaUR method therefore has value to the broader community. I think this study is a good first-step towards opening doors for the use of LLMs as fits of human data which would inspire further work in the space going forwards.

I am particularly excited by the generalization results the authors demonstrate in Section 3.4. It would be intriguing to see how performance scales when jointly training CENTaUR on many more human behavior datasets. I believe the methodology the authors introduce in the work, and the fact that they will release their code (and nicely, are using an already open-source model, in contrast to the oft used GPT series) should set their framework up well for extension.

I also think the inclusion of a random effects component in the finetuning layer to capute individual differences is quite a nice (and seemingly novel? To my understanding?) idea; I would encourage the authors to expand on this further as a contribution, if they indeed believe the method is more generalizable and important for extending LLMs as cognitive models.

I appreciate the authors’ rigour in their qualitative investigations (though I believe the results could be pushed on even further; see Weaknesses). I also appreciate the authors’ couching the broader implications of their work in the Discussion; the authors do a good job of cautioning against too broad a take on their findings of these models *as* cognitive models. Some of this could be brought further to the Intro (see below)).

**Weaknesses:**

While I think the authors do a good job in couching the generality of their method and limitations in the Discussion, I think this could be set-up further in the Introduction. At times, I feel the motivation was a bit confused. There is a difference, in my opinion, between seeking to bridge the gap between out-of-the-box language models and fits to behavioral data, versus actually using LLMs *as* cognitive models to *predict and study* human behavior. I believe this work nicely supports and expands on the former, and teases the possibility of the latter – but the results at present do not fully convince me that these models are ready yet, or should, serve as stimuli-computable models of human behavior.

Expanding on the above, it’s not clear to me that CENTaUR is better than BEAST (Fig 1c); yes, the NLL is objectively lower – but what about error bars? Is it significantly lower? The authors do a good job of highlighting qualitative strengths of CENTAuR, but it would be nice for a deeper dive on the gaps between CENTaUR and human performance as well? For instance, could the authors construct the inverse of Fig 5? What do examples of cases where CENTaUR did *not* match humans (but perhaps BEAST / hybrid did?) The authors note as well that there are 8 out of 60 individual participants not well-captured by the

More broadly, it would be nice if the authors could include error bars in their results as much as possible (it should be feasible to obtain these, given the authors used 100-fold CV?). Further, why are the more classical cognitive models not included in the plots in 2a and b?

As a minor stylistic weakness: I did not find it helpful to read the raw NLL values in the text. I think it’s best to keep these in the figures/tables and discuss the trends. It’s hard to tell what counts as a large or small gap between NLL; again, error bars would be helpful.

**Questions:**

Most of the hesitation in my final score stems from the Weaknesses above; if the authors are able to conduct a deeper dive into the potential gaps between CENTaUR and human performance (not just emphasizing the goodness of the fits) and further clarify the performance of CENTaUR relative to the existing cognitive baselines / broader motivation of these models as (replacement?) predictive cognitive models, I am open to raising my score. For instance, I believe that the authors resoundedly demonstrate that fine-tuning improves on LLaMA for fitting human behaviour, and this is a nice contribution; but I don't think the authors are yet ready to present their model as a stand-alone cognitive model. I realize the authors couch the limitations of this, but I think it could be further improved (per notes above).

I raise a few other points which I believe are important and/or could add to the paper, but these are worth addressing only if the authors have time.

- This is not so much a question, but would be nice for the authors to comment on (and perhaps include in the Discussion): The choices13k dataset appears to have been released in 2019? As such, it’s possible that the models were trained over this data (LLaMA performance is poor out-of-the-gate, so perhaps it is less of an issue; but it would be good for the authors to comment on as it could impact performance, and broadly a nuance in leveraging these models as cognitive models). One idea for the authors to begin to explore this is by repeating the prompt variation in Appendix A.4, but in the training set as well.
- Was there any interesting visible structure in the embeddings learned? Have the authors run any kind of visualization over the embeddings (tSNE, etc) and perhaps looked into any possible structure there which could inform why some participants rather than others are not captured by CENTaUR? This is not necessary and the lack of such an investigation does not lower my score; but could be an exploration to strengthen the work.
- Minor note: it would be nice if the authors could visualize the choice curves on the same graph. For instance, it looks like CENTaUR is much sharper in its curve than humans in Fig 4 (top row)? But this is a bit hard to cross-compare when the graphs are side-by-side.

---

> ### Author Response · Authors · 2023-11-16
> **Response to Reviewer donP**
>
> We thank the reviewer for their thoughtful and constructive feedback. The provided suggestions helped us to further improve our paper. We have incorporated them as described below.
>
> > While I think the authors do a good job in couching the generality of their method and limitations in the Discussion, I think this could be set-up further in the Introduction. [...] I believe this work nicely supports and expands on the former, and teases the possibility of the latter – but the results at present do not fully convince me that these models are ready yet, or should, serve as stimuli-computable models of human behavior.
>
> We agree with the reviewer’s interpretation of results and thank them for their feedback. We have updated the end of our introduction to better reflect this sentiment:
>
> **Taken together, our work demonstrates that it is possible to align large language models with human behavior by finetuning them on data from psychological experiments. We believe that this could open up new opportunities to harvest the power of large language models to inform our theories of human cognition in the future.**
>
> > Expanding on the above, it’s not clear to me that CENTaUR is better than BEAST (Fig 1c); yes, the NLL is objectively lower – but what about error bars? Is it significantly lower?
>
> We thank the reviewer for bringing up this question. There are two main ways of conducting significance tests in our setting: (a) treating outcomes of individual participants as samples from a random variable or (b) treating outcomes for runs with different seeds as samples from a random variable.
>
> (a) can only be done for the horizon task data as data from individual participants is not available for the choices13k data set. That being said, we have added an additional analysis regarding (b) to our paper. We have repeated our model fitting procedure 5 times and performed significance tests on the resulting data. In general, we found very little variation across seeds. We have added the following information to our updated PDF:
>
> **To ensure that our fitting procedure is robust to random noise, we repeated this analysis with five different seeds. CENTaUR's results were almost identical across seeds with an average negative log-likelihood of 48002.3 (SE = 0.02) for the choices13k data set and an average negative log-likelihood of 25968.6 (SE = 0.02) for the horizon task. In both cases, these values were significantly lower than those of the domain-specific models (p<0.001).**
>
> > The authors do a good job of highlighting qualitative strengths of CENTAuR, but it would be nice for a deeper dive on the gaps between CENTaUR and human performance as well? For instance, could the authors construct the inverse of Fig 5? What do examples of cases where CENTaUR did not match humans (but perhaps BEAST / hybrid did?)
>
> We found this suggestion very interesting and have therefore conducted the corresponding analysis:
>
> **We have also conducted an inverse of the analysis presented in Section 3.5. For this analysis, we inspected the data points with the lowest log-likelihood difference between CENTaUR and the domain-specific model. These points correspond to cases where the domain-specific model accurately predicted human behavior but CENTaUR did not. Figure 8 shows three examples for each paradigm.**
>
> The results can be found in our updated PDF in Appendix A.6. Unfortunately, we are not able to identify any underlying pattern in these data points.
>
> > This is not so much a question, but would be nice for the authors to comment on (and perhaps include in the Discussion): The choices13k dataset appears to have been released in 2019? As such, it’s possible that the models were trained over this data.
>
> We thank the reviewer for pointing this out. It is, in principle, possible that the data sets are part of CommonCrawl which is used to train LLaMA. However, we believe that this is unlikely given that the raw data is stored as .csv and .mat files. Further evidence for this conjecture comes from the observation that non-finetuned LLaMA does not predict human choices well.
>
> > Was there any interesting visible structure in the embeddings learned?
>
> We have indeed conducted a few basic embedding visualizations. We have now included them in our updated PDF under Appendix A.7. In particular, we write:
>
> **We performed a simple t-SNE (Van der Maaten & Hinton, 2008) analysis to investigate the information contained in the LLaMA embeddings. Figure 7 shows the results of this analysis. We found that the different tasks can be separated in the t-SNE space and that the embeddings for the experiential-symbolic task lie in between those of the two individual tasks. While the t-SNE space also displays some structure with regard to the difference in values between both options, the overall picture is less clear in this case.**

---

> > ### Comment · Reviewer_donP · 2023-11-16
> > **Thank you for your response!**
> >
> > Dear Authors,
> >
> > Thank you for your clear and helpful rebuttal to my response and the other Reviewers! I feel comfortable now recommending the paper for Acceptance and have increased my score accordingly.

---

### Official Review · Reviewer_nBjG · 2023-10-24

**Soundness:** 3 good
**Presentation:** 4 excellent
**Contribution:** 3 good
**Rating:** 8
**Confidence:** 5

**Summary:**

This work presents a novel cognitive model that uses embeddings from an LLM (LLaMa), together with a regression model, to predict human decision-making behavior. The model outperforms both the pre-trained LLM and domain-specific cognitive models, accounts for individual differences, and can generalize to a holdout task.

**Strengths:**

- This work presents an interesting and novel approach to cognitive modeling that outperforms domain-specific cognitive models.
- The model is shown to be capable of generating qualitative cognitive insights, in addition to superior quantitative performance.
- The model accounts for individual differences.
- The model generalizes to a novel task.
- The paper includes some interesting discussion of the broader implications of this approach for cognitive science.

**Weaknesses:**

- I am not sure if this is a weakness per se, but the work is primarily oriented toward cognitive science. It may be better suited to a more cog-sci oriented venue. However, I think the work generally makes a strong contribution and would be happy for it be published at ICLR.
- My primary substantive concern is that the model is only evaluated on publicly available datasets. Do the authors know whether this data is included in LLaMa's pretraining data? I'm not entirely certain, but given the open-source nature of the model I think it should be possible to determine this. This seems like an important factor for considering how generalizable the approach will be to new datasets.
- For the holdout task, is there an appropriate domain-specific model with which to compare CENTaUR?

**Questions:**

- Have the authors considered whether a more powerful LLM (e.g. GPT-3 or GPT-4) might perform better on these tasks without fine-tuning, or carried out any such evaluation? My sense is that LLaMa is not as effectively instruction-tuned as these other models, and thus performs poorly in the zero-shot setting, which would explain the need for fine-tuning. But I wonder whether a more effectively instruction-tuned model might perform better 'out of the box' (I'm not suggesting that the authors need to perform this evaluation, just curious to hear their thoughts).
- Have the authors tried to train LLaMa through in-context learning instead of fine-tuning? If this were effective, it might be more useful in settings where training data is limited (as is more generally the case in cognitive science).
- Is CENTaUR an acronym, and if so what does it stand for?

---

> ### Author Response · Authors · 2023-11-16
> **Response to Reviewer nBjG**
>
> We thank the reviewer for their generous review. We clarify the remaining open questions below.
>
> > I am not sure if this is a weakness per se, but the work is primarily oriented toward cognitive science. It may be better suited to a more cog-sci oriented venue. However, I think the work generally makes a strong contribution and would be happy for it be published at ICLR.
>
> We agree that our work caters to a cognitive science audience. However, ICLR’s call for papers specifically mentions “applications to neuroscience & cognitive science” under relevant topics, and we believe our work is a perfect example of this.
>
> > My primary substantive concern is that the model is only evaluated on publicly available datasets. Do the authors know whether this data is included in LLaMa's pretraining data? I'm not entirely certain, but given the open-source nature of the model I think it should be possible to determine this. This seems like an important factor for considering how generalizable the approach will be to new datasets.
>
> We thank the reviewer for pointing this out. It is, in principle, possible that the data sets are part of CommonCrawl which is used to train LLaMA. However, we believe that this is unlikely given that the raw data is stored as .csv and .mat files. Further evidence for this conjecture comes from the observation that non-finetuned LLaMA does not predict human choices well.
>
> > For the holdout task, is there an appropriate domain-specific model with which to compare CENTaUR?
>
> The original paper that introduced this experimental paradigm did not suggest a domain-specific model. While we could potentially come up with one, we think that this is out-of-scope for the current project.
>
> > Have the authors considered whether a more powerful LLM (e.g. GPT-3 or GPT-4) might perform better on these tasks without fine-tuning, or carried out any such evaluation?
>
> Thank you for this suggestion. We have included GPT-4 as a baseline in our analysis based on this feedback. In general, we find that GPT-4 predicts human behavior better than LLaMA but not to the level of CENTaUR. Note that we had to perform this analysis using accuracy as a metric since GPT-4 does not provide us with log probabilities. We have added the following text to our paper:
>
> **Figure 6 contains a validation of our main analysis using accuracy as a measurement of goodness-of-fit. We again found that CENTaUR was the best performing model, achieving accuracies of 65.18% for the choices13k data set and 83.5% for the horizon task. We also included GPT-4 (OpenAI, 2023) as a baseline in this analysis. GPT-4 performed worse when compared to CENTaUR for both the choices13k data set (59.72%) and the horizon task (80.3%) but better than LLaMA without any finetuning.**
>
> The resulting plot can be found in Appendix A.3 in our updated PDF.
>
> > Have the authors tried to train LLaMa through in-context learning instead of fine-tuning? If this were effective, it might be more useful in settings where training data is limited (as is more generally the case in cognitive science).
>
> We have not considered this for the present project but it would certainly be an interesting direction for future research.
>
> > Is CENTaUR an acronym, and if so what does it stand for?
>
> It is not an acronym. The capitalization was chosen in a way that matches that of LLaMA.

---

> > ### Comment · Reviewer_nBjG · 2023-11-22
> > **Response to rebuttal**
> >
> > Thanks very much to the authors for these responses and clarifications. I enthusiastically support the paper's acceptance.

---

### Official Review · Reviewer_sVTi · 2023-10-31

**Soundness:** 3 good
**Presentation:** 4 excellent
**Contribution:** 3 good
**Rating:** 8
**Confidence:** 5

**Summary:**

This paper studies the problem of using large language models to simulate human results in behavioral experiments, with a focus on decision-making tasks. The method is intuitive and effective: linear probing of a llama model with human behavioral experiment data (i.e., extract latent embeddings and feed to a linear classifier to predict the judgment). Experiment results show such a fine-tuned-based approach can well fit human behavioral experiment distribution and generalize to hold-out tasks. This work shed light on some alternative approaches besides Bayesian models for predicting human decision-making results.

**Strengths:**

- This paper is well written, with a clear motivation of using large language models to simulate human behaviors (or at least binary choice results on two types of decision-making tasks). The detailed implementation: extract embeddings and then do a linear probing is good enough for a scalable method.

- I like the idea of using large language models for a proxy model for analyzing human behavior. The crux is how to prob or design proper experimental methods (analogous to methods developed in experiment psychology). The proposed fine-tuning method performs well on behavior prediction tasks.

- The main results are good, showing good predictive matches to human participants compared with other common methods. The hold-out tests also confirm the validity of using large language models as a universal cognitive model for behavioral tests.

**Weaknesses:**

- Although using open-sourced models (e.g., llama) is a good choice, the most powerful models to date (including instruction fine-tuned ones)  are not tested as a baseline method (e.g., few-shot evaluations on GPT-3.5/GPT-4/PaLM-2/Claude/instruction-finetuned llama 2 variants), it is suggested that some of those models can also demonstrate human-like behaviors in some human decision making tasks, through prompting or few-shot evaluations [A]. The few-shot method might be done by prompting some of the training examples in the CENTaUR method. So, it might be helpful to prove that CENTaUR is better than other prompting-based methods on more powerful LLMs.

- Missing some discussions to previous works using LLMs to simulate and replicate behavior study results: [A-E]. Some similar aspects for using LLM in decision-making evaluations have already been proposed by Aher et al. [A].

- Minor problems (only suggestions, not affecting the score): please use vectorized graphics (Figure 6 is good, but Figures 1-5 look blurry).

ref:

[A]. Aher, G. V., Arriaga, R. I., & Kalai, A. T. (2023). Using large language models to simulate multiple humans and replicate human subject studies. In ICML 2023

[B]. Jiang, G., Xu, M., Zhu, S. C., Han, W., Zhang, C., & Zhu, Y. (2023). Evaluating and Inducing Personality in Pre-trained Language Models. In NeurIPS 2023

[C]. Frank, M. C. (2023). Large language models as models of human cognition. PsyArXiv

[D]. Shiffrin, R. and Mitchell, M. (2023). Probing the psychology of AI models. PNAS

[E]. Demszky, D., Yang, D., Yeager, D.S. et al. (2023). Using large language models in psychology. Nat Rev Psychol. *this one is not within the three months before the submission, so it is perfectly fine not to mention it :), just as a suggestion*

**Questions:**

See weaknesses. I'm open to discussing and increasing my score. The score I'm willing to give now is ~7, but this year's ICLR only has 6 and 8 options :). Moreover, is there any statistical analysis (e.g., significance tests) for comparing the correlation between human and CENTaUR predictions?

---

> ### Author Response · Authors · 2023-11-16
> **Response to Reviewer sVTi**
>
> We thank the reviewer for the thorough and helpful feedback. We have addressed the three weaknesses outlined by the reviewer and doing so helped us to further improve our paper.
>
> > Although using open-sourced models (e.g., llama) is a good choice, the most powerful models to date (including instruction fine-tuned ones) are not tested as a baseline method (e.g., few-shot evaluations on GPT-3.5/GPT-4/PaLM-2/Claude/instruction-finetuned llama 2 variants), it is suggested that some of those models can also demonstrate human-like behaviors in some human decision making tasks, through prompting or few-shot evaluations [A]. The few-shot method might be done by prompting some of the training examples in the CENTaUR method. So, it might be helpful to prove that CENTaUR is better than other prompting-based methods on more powerful LLMs.
>
> We thank the reviewer for raising this point. We have included GPT-4 as a baseline in our analysis based on this feedback. In general, we find that GPT-4 predicts human behavior better than LLaMA but not to the level of CENTaUR. Note that we had to perform this analysis using accuracy as a metric since GPT-4 does not provide us with log probabilities. We have added the following text to our paper:
>
> **Figure 6 contains a validation of our main analysis using accuracy as a measurement of goodness-of-fit. We again found that CENTaUR was the best performing model, achieving accuracies of 65.18% for the choices13k data set and 83.5% for the horizon task. We also included GPT-4 (OpenAI, 2023) as a baseline in this analysis. GPT-4 performed worse when compared to CENTaUR for both the choices13k data set (59.72%) and the horizon task (80.3%) but better than LLaMA without any finetuning.**
>
> The resulting plot can be found in Appendix A.3 in our updated PDF.
>
> > Missing some discussions to previous works using LLMs to simulate and replicate behavior study results: [A-E]. Some similar aspects for using LLM in decision-making evaluations have already been proposed by Aher et al. [A].
>
> Thanks a lot for pointing us to these references. We have included all of them at the appropriate places:
>
> Recently, researchers even started to wonder whether large language models can replace human participants in experimental studies (Dillion et al., 2023), **i.e., whether they can be used as proxies for human behavior (Aher et al., 2023; Jiang et al., 2022).**
>
> To summarize, large language models are an immensely powerful tool for studying human behavior **(Demszky et al., 2023; Frank, 2023b).**
>
> Previous work has shown that these models can even successfully navigate when they are placed into classic psychological experiments (Binz & Schulz, 2023; **Shiffrin & Mitchell, 2023;** Coda-Forno et al., 2023; Dasgupta et al., 2022; Hagendorff et al., 2022).
>
> > Minor problems (only suggestions, not affecting the score): please use vectorized graphics (Figure 6 is good, but Figures 1-5 look blurry).
>
> We have updated all figures with their vectorized equivalents (see updated PDF).
>
> > Moreover, is there any statistical analysis (e.g., significance tests) for comparing the correlation between human and CENTaUR predictions?
>
> We thank the reviewer for bringing up this question. There are two main ways of conducting significance tests in our setting: (a) treating outcomes of individual participants as samples from a random variable or (b) treating outcomes for runs with different seeds as samples from a random variable.
>
> (a) can only be done for the horizon task data as data from individual participants is not available for the choices13k data set. In fact, we have already done an analysis like this for the horizon task data: the random-effects model selection procedure from Section 3.3 can be considered as a fancy Bayesian version of a significance test (it gives the probability that a model is better than the considered alternatives).
>
> Furthermore, we have added an additional analysis regarding (b) to our paper. We have repeated our model fitting procedure 5 times and performed significance tests on the resulting data. In general, we found very little variation across seeds. We have added the following information to our updated PDF:
>
> **To ensure that our fitting procedure is robust to random noise, we repeated this analysis with five different seeds. CENTaUR's results were almost identical across seeds with an average negative log-likelihood of 48002.3 (SE = 0.02) for the choices13k data set and an average negative log-likelihood of 25968.6 (SE = 0.02) for the horizon task. In both cases, these values were significantly lower than those of the domain-specific models (p<0.001).**

---

> > ### Comment · Reviewer_sVTi · 2023-11-16
> >
> > Thank the authors' response. Most of my concerns are resolved, and I'll increase my rating by one.

---

### Official Review · Reviewer_RhDW · 2023-11-02

**Soundness:** 3 good
**Presentation:** 3 good
**Contribution:** 3 good
**Rating:** 8
**Confidence:** 5

**Summary:**

The authors propose to fine-tune large language models (LLM) on real human decision making behavior / tasks from psychology to potentially better align them with people and provide better predictions of behavior. They find that fine-tuning is largely successful on two such tasks and allow for at least some generalization to a third hold out task.

**Strengths:**

I found this paper really interesting and worthwhile. While I suspect that there is much more work to be done in assessing the results, the main result is clear and fascinating: LLMs can quickly adapt to predict specific human behaviors. Perhaps most interesting is that this only requires linear regression. While this isn't really fine-tuning in the typical sense and could be considered a limitation (the authors could have trained a single layer or so, but with LLMs this is not easy by any means), I find it useful to know that only a simple transformation is needed. In fact, one could argue that the model didn't really have to learn anything about people. Everything the model needs appears to be located in the final input representation. One could further argue that this implies that LLMs already know how to emulate a diverse range of behavior, but that prompting is not the best way to test for it. In that sense, I think the primary contribution is strong. I don't think the authors needed to name their linear model, as the contribution is empirical and not methodological in some sense, but that's not a weakness given the aim of the paper.

**Weaknesses:**

The main weakness in my view is the difficulty in comparing to past work in the relevant domains and which are cited in the work. Past work appears to use different metrics, different splits of the data, and different baseline models. For example, BEAST does not appear to be the best or only relevant baseline, which is also usually evaluated using mean squared error. There is also a history of work behind the choices13k dataset with machine learning methods that the authors don't review. The authors also don't provide more interpretable metrics such as accuracy. One option to fix this would be make better comparisons using a larger set of relevant models, while another could be to focus on the success of fine-tuning more generally, which is already interesting, and less on benchmarks.

More generally, the claims of the paper could be toned down a bit. The space of human behavior is massive, and success in fine-tuning to extremely narrow ranges of behavior does not guarantee any particular success rate with other kinds. LLMs already fall short in matching human intelligence in some respects, and thus can't mimic such behavior. This may even be true or more nuanced "non-intelligent" behavior as well.

**Questions:**

--- What is the practical improvement (e.g., accuracy?) of fine-tuning?
--- Can BEAST be fine-tuned as well?
--- How does this compare to more typical feedforward neural networks with no pretraining.

---

> ### Author Response · Authors · 2023-11-16
> **Reponse to Reviewer RhDW**
>
> We thank the reviewer for their positive and constructive feedback. We have updated our paper based on several suggestions by the reviewer as outlined below.
>
> > The main weakness in my view is the difficulty in comparing to past work in the relevant domains and which are cited in the work. Past work appears to use different metrics, different splits of the data, and different baseline models. For example, BEAST does not appear to be the best or only relevant baseline, which is also usually evaluated using mean squared error. There is also a history of work behind the choices13k dataset with machine learning methods that the authors don't review.
>
> We have opted to measure goodness-of-fit on the choices13k data set using the negative log-likelihood (instead of the MSE which is commonly used for this data set) because this allows us to use consistent metrics across tasks. We agree with the reviewer that there are other plausible baselines. To highlight this, we have added the following footnote:
>
> **There are a number of other domain-specific models that could be used as baselines instead (Bourgin et al., 2019; He et al., 2022; Zhang & Yu, 2013; Wilson et al., 2014). While the ones we selected are good representatives of the literature, future work should strive to conduct a more exhaustive comparison.**
>
> The footnote also refers to other work on the choices13k data set. We are happy to add additional references here if there are any specific suggestions.
>
> > The authors also don't provide more interpretable metrics such as accuracy.
>
> Thanks a lot for this suggestion. We have replicated our main analysis using accuracy as a metric. The results mirror those obtained using negative log-likelihoods. We have added the following text to our paper:
>
> **Figure 6 in Appendix A.3 further corroborates these results using accuracy as a measurement of goodness-of-fit.**
>
> and the following to the Appendix:
>
> **Figure 6 contains a validation of our main analysis using accuracy as a measurement of goodness-of-fit. We again found that CENTaUR was the best performing model, achieving accuracies of 65.18% for the choices13k data set and 83.5% for the horizon task.**
>
> The resulting plot can be found in Appendix A.3 in our updated PDF.
>
> > More generally, the claims of the paper could be toned down a bit. The space of human behavior is massive, and success in fine-tuning to extremely narrow ranges of behavior does not guarantee any particular success rate with other kinds.
>
> We agree with the reviewer’s assessment and have been careful to not make such a claim. For example, we write that:
>
> We have presented preliminary results in this direction, showing that a model finetuned on two tasks can predict human behavior on a third. However, we believe that our current results only hint at the potential of this approach.
>
> and:
>
> However, it is important to mention that the present paper still falls short of this goal.
>
> Furthermore, we have modified the ending of our introduction to tone down some of its claims a bit:
>
> We show that this approach can be used to create models that describe human behavior better than traditional cognitive models. We verify our result through extensive model simulations, which confirm that finetuned language models indeed show human-like behavioral characteristics. Furthermore, we find that embeddings obtained from such models contain the information necessary to capture individual differences. Finally, we highlight that a model finetuned on two tasks is capable of predicting human behavior on a third, hold-out task. **Taken together, our work demonstrates that it is possible to align large language models with human behavior by finetuning them on data from psychological experiments. We believe that this could open up new opportunities to harvest the power of large language models to inform our theories of human cognition in the future.**

---

> > ### Comment · Reviewer_RhDW · 2023-11-16
> > **Additional question**
> >
> > Thank you. 65.18% seems low compared to the baseline for choices13k (~85%). Can you comment on this?

---

> > > ### Author Response · Authors · 2023-11-17
> > >
> > > We assume that the ~85% come from the Peterson et al. Science paper. The discrepancy here arises from the fact that Peterson et al. report decision preference accuracy (“i.e., the proportion of problems in which the model prediction for P(A) is >0.5 when the observed proportions are also >0.5.”) whereas we report plain accuracy. When we compute the decision preference accuracy for CENTaUR we find values that are more closely aligned with those reported by Peterson et al. (around 82%). The small differences likely come from differences in the evaluation procedure, e.g., different splits of the data.
> > >
> > > We made this choice for two reasons: 1. it allows us to also include GPT-4 in the comparison (requested by several other reviewers), and 2. we can use the same metric for the horizon task data. For both GPT-4 and the horizon task data, one can not compute decision preference accuracy.
> > >
> > > We are glad the reviewer pointed out this issue as it has the potential to cause misunderstanding. We will add the following footnote to our camera-ready version to prevent this:
> > >
> > > **Note that the accuracy differs from the decision preference accuracy reported by Peterson et al. (2021). The decision preference accuracy is defined as the proportion of problems in which the model prediction for P(A) is >0.5 when the observed proportions are also >0.5. CENTaUR achieves a decision preference accuracy of 81.9%.**

---

> > > > ### Comment · Reviewer_RhDW · 2023-11-22
> > > > **Acknowledgement of rebuttal**
> > > >
> > > > Thanks for clarifying. I continue to recommend acceptance.

---

### Meta-Review · Area_Chair_RFXe · 2023-12-12

**Metareview:**

The submission demonstrates that linear probing of a large language model, LLaMA, on psychological decision-making data improves alignment with human behavior, allowing for accurate predictions on held-out decision-making data, though with limited generalization to new prompt structures. The technique is simple and less domain-specific than prior psychological models, and the results will be intriguing to both machine learning practitioners and psychological scientists.

**Justification For Why Not Higher Score:**

scope of the evaluation is two constrained decision-making tasks; lack of robustness to changes in prompt structure is not investigated in detail nor addressed

**Justification For Why Not Lower Score:**

timely topic; straightforward contribution

---

### Decision · Program_Chairs · 2024-01-16

Accept (poster)